# WormPose: Image synthesis and convolutional networks for pose estimation in *C. elegans*

**Laetitia Hebert**[1], **Tosif Ahamed**[1,2], **Antonio C. Costa**[3], **Liam O'Shaughnessy**[3], **Greg J. Stephens**[1,3]*

**1** Biological Physics Theory Unit, OIST Graduate University, Onna, Japan, **2** Lunenfeld-Tanenbaum Research Institute, University of Toronto, Toronto, Canada, **3** Department of Physics & Astronomy, Vrije Universiteit Amsterdam, Amsterdam, Netherlands

* g.j.stephens@vu.nl

## Abstract

An important model system for understanding genes, neurons and behavior, the nematode worm *C. elegans* naturally moves through a variety of complex postures, for which estimation from video data is challenging. We introduce an open-source Python package, Worm-Pose, for 2D pose estimation in *C. elegans*, including self-occluded, coiled shapes. We leverage advances in machine vision afforded from convolutional neural networks and introduce a synthetic yet realistic generative model for images of worm posture, thus avoiding the need for human-labeled training. WormPose is effective and adaptable for imaging conditions across worm tracking efforts. We quantify pose estimation using synthetic data as well as N2 and mutant worms in on-food conditions. We further demonstrate WormPose by analyzing long ($\sim$ 8 hour), fast-sampled ($\sim$ 30 Hz) recordings of on-food N2 worms to provide a posture-scale analysis of roaming/dwelling behaviors.

## Author summary

Recent advances in machine learning have enabled the high-resolution estimation of bodypoint positions of freely behaving animals, but manual labeling can render these methods imprecise and impractical, especially in highly deformable animals such as the nematode *C. elegans*. Such animals also frequently coil, resulting in complicated shapes whose ambiguity presents difficulties for standard pose estimation methods. Efficiently solving coiled shapes in *C. elegans*, exhibited in a variety of important natural contexts, is the primary limiting factor for fully automated high-throughput behavior analysis. WormPose provides pose estimation that works across imaging conditions, naturally complements existing worm trackers, and harnesses the power of deep convolutional networks but with an image generator to automatically provide precise image-centerline pairings for training. We apply WormPose to on-food recordings, finding a near absence of deep $\delta$-turns. We also show that incoherent body motions in the dwell state, which do not translate the worm, have been misidentified as an increase in reversal rate by previous, centroid-based methods. We expect that the combination of a body model and image

**Data Availability Statement:** The data is available here: https://wormpose.unit.oist.jp.

**Funding:** We acknowledge funding from the Vrije Universiteit Amsterdam and OIST Graduate

University. The funders had no role in study design, data collection and analysis, decision to publish, or preparation of the manuscript.

**Competing interests:** The authors have declared that no competing interests exist.

synthesis demonstrated in WormPose will be both of general interest and important for future progress in precise pose estimation in other slender-bodied and deformable organisms.

## Introduction

All animals, including humans, reveal important and subtle information about their internal dynamics in their outward configurations of body posture, whether these internal dynamics originate from gene expression [1], neural activity [2], or motor control strategies [3]. Estimating and analyzing posture and posture sequences from high-resolution video data is thus a general and important problem, and the basis of a new quantitative approach to movement behavior (for reviews see e.g. [4, 5]).

The roundworm *C. elegans*, important on its own as a model system (see e.g. [6]), provides an illustrative example, where "pose" can be identified as the geometry of the centerline extracted from worm images [7]. Even with a relatively simple body plan, identifying the centerline can be challenging due to coiling and other self-occluded shapes, Fig 1. These shapes occur in important behaviors such as an escape response [8, 9], among mutants [10] and are a yet unanalyzed component in increasingly copious and quantitative recordings such as the Open Worm Movement Database [11].

Classical image skeletonization methods can be used to identify the worm centerline for non-overlapping shapes [7] and are employed in widely-used worm trackers because of their simplicity and speed. For coiled or self-overlapping postures, more advanced statistical models combine image features such as edges with a model of the worm's centerline [10, 12–15]. However, such image features are not always visible and are not robust to changes in noise or brightness, often requiring data-specific engineering which reduces portability. Another recent technique utilizes an optimization algorithm by searching for image matches in the "eigen-worm" posture space [9], but is limited in efficacy by the slow nature of multi-dimensional image search and by the low resolving power of a comparison metric, which uses only a binary version of the raw image.

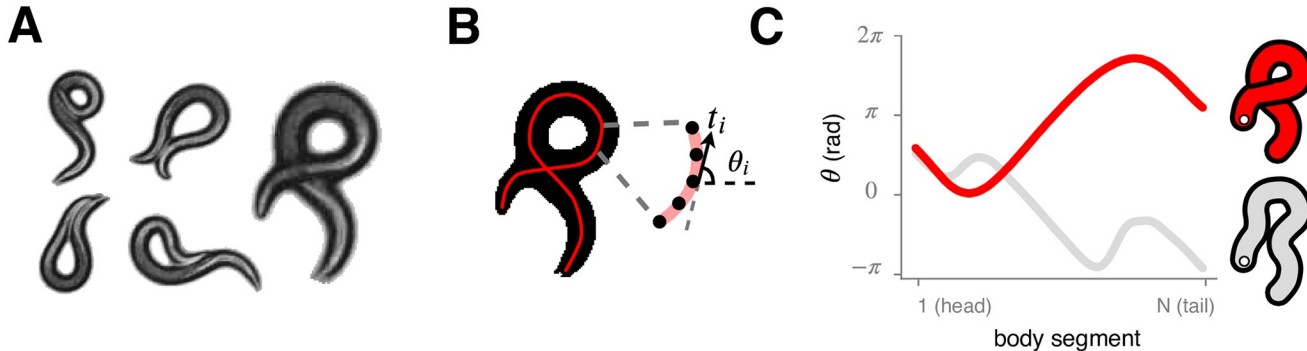

**Fig 1. The nematode *C. elegans* naturally exhibits a variety of coiled shapes which challenge the determination of the centerline posture, a fundamental component for quantitative behavioral understanding.** (A) An exemplar collection of images displaying coiled shapes. (B) Instantaneous worm pose encoded as the centerline curve parameterized by tangent angles $\boldsymbol{\theta} = (\theta_1, \ldots \theta_i, \ldots \theta_N)$ ordered from head to tail. (C) Standard image processing techniques extract the centerline by morphological operations or image features analysis and have not been able to differentiate solutions with very different centerlines (red, grey) that occur with coiled posture. The correct centerline (red) can be determined by close visual inspection (A), however high-throughput analysis necessitates a pose estimation algorithm which is robust to fluctuations in brightness, blur, noise, and occlusion.

With the ability to extract complex visual information about articulated objects, methods built from convolutional neural networks (CNNs) offer a new, promising direction. CNNs are the foundation for recent, remarkable progress in markerless body point tracking [16–18], including worm posture [19]. However, intensive labeling requirements by human annotators, even if assisted by technology [20], as well as the ambiguity of which or exactly how many points to label, offer a barrier to the usefulness of CNNs in posture tracking and beyond. Body point marking is challenging in the case of worm images where the annotation task is to label enough points along the worm body to reconstruct the posture. While human annotators can quickly pinpoint the extremities of the worm body, other landmarks are less obvious. In some recordings, it is even difficult to distinguish the worm head from the tail, which makes the labeling error-prone and imprecise. Furthermore, the labeling is specific to the recording conditions and can be hard to generalize across changes in resolution, organism size, background, illumination, and to rare posture configurations not specifically isolated.

We describe an algorithm, WormPose, for pose estimation in *C. elegans* containing two principal advances: (1) We create a model of worm shape probabilities (a generative model) which we combine with a new technique for producing synthetic but realistic worm images. These images are used for network training, thus circumventing the difficulty and ambiguity of human labeling, and can be easily adapted to different imaging conditions (2) We develop a CNN to reliably transform worm images to a centerline curve. We demonstrate our approach using on-food behavior of N2 and mutant worms and use our results to provide a new posture-scale analysis of roaming and dwelling behavioral states. Compared to prior work, the CNN at the core of our algorithm results in dramatically increased computational speed, providing new opportunities for scientific exploration.

## Design & implementation

### Data requirements

Our focus is on resolving coiled, overlapping, blurred, or other challenging images of a single worm. We assume that the input data consists of videos of a single moving worm and that most of the non-coiled frames are analyzed beforehand, for example by Tierpsy tracker [21]. For each (non-coiled) frame, we require the coordinates of equidistant points along the worm centerline, ordered from head to tail, and the worm width for the head, midbody, and tail (defined in [22]). We also use the recording frame rate. WormPose 1.0 does not detect the head of the worm, so we also expect that the labeled frames provide the head-tail position at regular intervals throughout the video. Head-tail positions are an included output of the Tierpsy tracker.

### Processing worm images

From a dataset described as above, we process worm images to focus on the worm object of interest. Broadly, we first segment the worm in the image and set all non-worm pixels to a uniform color. Then we either crop or extend to create a square image of uniform size with the worm in the center, cleaned of noise and non-worm objects.

The specific process of segmenting a single worm in an image can be adapted to each recording condition. For concreteness, we provide a simple OpenCV [23] implementation that is sufficient for most videos of the Open Worm Movement Database [11]. Raw images from the video are first processed by a Gaussian Blur filter with a window size of 5 pixels, and then thresholded with an automatic Otsu threshold to separate the background and the foreground. The morphological operation "close" is applied to fill in the holes in the foreground image. We use a connected components function to identify the objects belonging to the

foreground. To focus on objects located at the center of the image, we crop the thresholded image on each side by an amount consisting of 15% of the size of the image. We isolate the largest blob in this cropped image as the worm object of interest. We calculate the background color as the average of the background pixels of the original image, and assign this background value to all pixels that do not belong to the worm object. All processed images are then either cropped or extended to be the same width and height, with the worm object set in the center. We set the default value for the processed image size as the average worm length of the biggest worm in the dataset, a size large enough to encompass all examples. Alternatively, the image size can be set by the user, and the images will be resized with linear interpolation, which is useful to speed up computation on large images. The minimum image size is $32 \times 32$ pixels.

## Generating worm shapes

We generate realistic worm shape through a Gaussian Mixture Model (GMM) Fig 2A, which we fit to a collection of resolved body postures obtained from previous analysis [7, 9]. We use the GMM as a simple generative model of body shapes that enables the sampling of an arbitrarily large training set for the network, with shapes that respect the overall correlations between body parts while generalizing to more complex postures. We parameterize worm

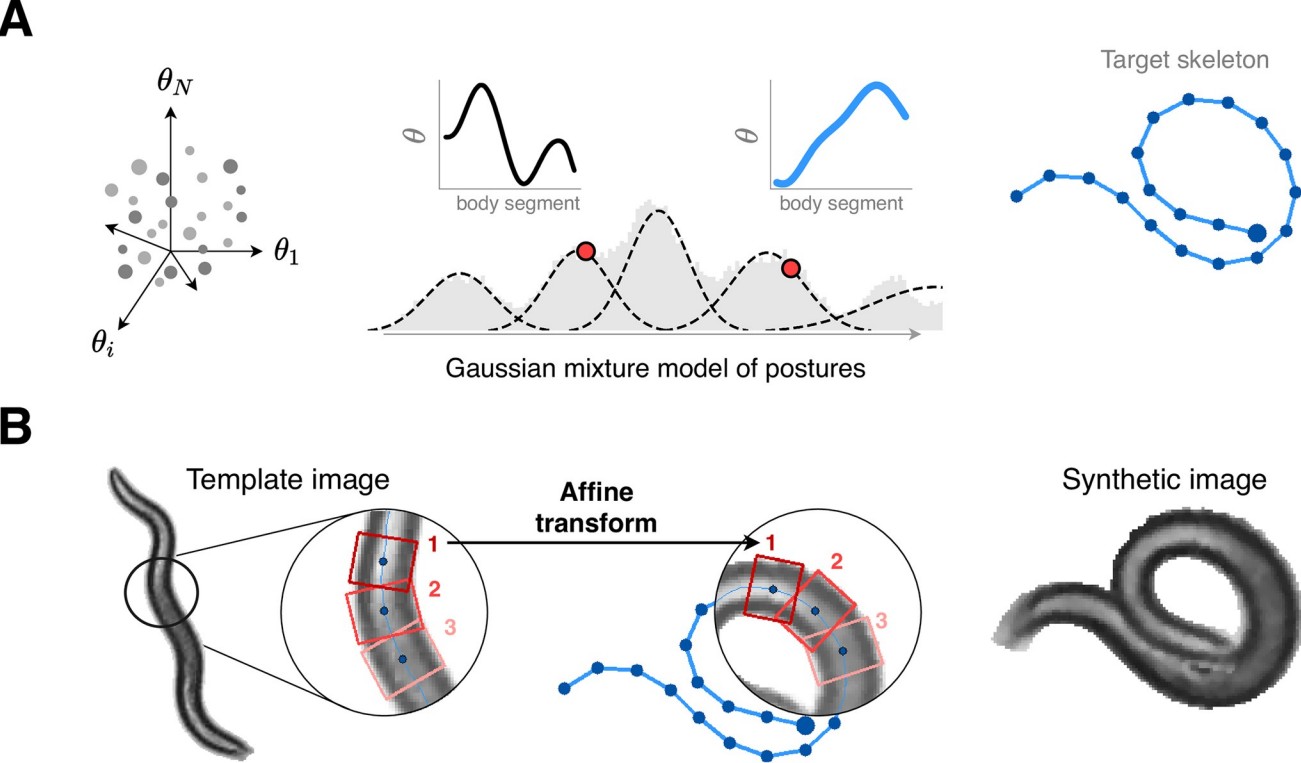

**Fig 2. We combine a generative model of worm posture with textures extracted from real video to create realistic yet synthetic images for a wide variety of naturalistic postures, including coils, thus avoiding the need for manually-annotated training data.** (A) We model the high-dimensional space of worm posture (left) by Gaussian mixtures (middle) constructed from a core set of previously analyzed worm shapes [9]. To each generated posture we add a global orientation (chosen uniformly between 0 and $2\pi$), and we randomly assign the head to one end of the centerline. (right) We use the resulting centerline (angle coordinates) to construct the posture skeleton (pixel coordinates). (B) We warp small rectangular pixel patches along the body of a real template image (left) to the target centerline (middle), producing a synthetic worm image (right). Overlapping pixels are alpha-blended to connect the patches seamlessly. Unwanted pixels protruding from the target worm body are masked and the background pixels are set to a uniform color. Finally, the image is cleaned of artifacts through a medium blur filter.

shape by a 100-dim vector of angles $\vec{\theta}$, formed by measuring the angle between 101 points equally spaced along the body's centerline. Uncoiled shapes were obtained using classical image tracking to extract $\vec{\theta}$ directly from images [7]. Coiled shapes were obtained in [9] by searching the lower-dimensional space of eigenworm projections ($d = 5$, obtained through Principal Component Analysis of the space of $\{\vec{\theta}\}$), to find the combination of eigenworm coefficients that best matches a given image and projecting these back into the $\vec{\theta}$ space. Using the classical image analysis results from [7] allows us to expand the space of possible $\vec{\theta}$ beyond the one captured by the first 5 eigenworms used in [9]. We use an equal population of coiled and uncoiled postures from N2 worms foraging off-food and sample uniformly according to the body curvature as measured by the third eigenworm projection, $a_3$. This yields a training set of $\sim 15000$ $\vec{\theta}$ vectors. We fit the GMM through an Expectation-Maximization algorithm which finds the set of $N$ Gaussian components that maximizes the likelihood (see e.g. [24]). The full model is parameterized by the mean and covariance of each Gaussian, and the weight associated with each Gaussian component. We assess the trade-off between model complexity and accuracy with Akaike's information criterion, which indicates that $N \sim 250 - 275$ components would be an appropriate choice, S1 Fig. We set $N = 270$ components for this manuscript, but $N$ can be tuned by the user according to the desired degree of variability in the generated worm shapes: larger $N$ decreases the variability (the GMM is closer to the underlying training set), while lower $N$ increases the variability in the obtain worm shapes. We train the GMM using `sklearn.mixture.GaussianMixture` from scikit-learn in Python [25].

## Generating synthetic images

We build a synthetic image generator to produce a worm image with a specific posture and with the same appearance as a reference image, Fig 2B. Such synthetic images have a similar appearance to real images processed as described above.

We exclusively use classical image processing techniques, including image warping and alpha blending, to effectively bend a known worm centerline from a reference image into a different posture. The reference image is typically of a non-overlapping worm, with its associated labeled features: (1) the skeleton as a list of $N_S$ coordinates $(S_x, S_y)$ equidistant along the centerline, and (2) the worm width at three body points: head, midbody and tail. To create a new synthetic image we first draw a centerline $\vec{\theta}$ of size 100 from the GMM worm shape generator. We produce target skeleton coordinates $\{S_x, S_y\}$ through the transformation

$$S_x(i + 1) = S_x(i) + dS \cos \theta_i \tag{1}$$

$$S_y(i + 1) = S_y(i) + dS \sin \theta_i$$

for $i = 1, 2 \ldots N_S$. The length element $dS$ is determined by dividing the worm length of the reference image by $N_S - 1$ and we set the origin by centering the skeleton in the middle of the target image, Fig 2A(right). If needed, the target skeleton is resampled to have the same number of points $N_S$ as the reference skeleton. We use the labeled width for the head, midbody and tail to calculate the worm width (in pixels) at all skeleton points $ww(i)$:

```
ww[0:head]=head_width
ww[head:midbody]=interp(head_width, midbody_width)
ww[midbody:tail]=interp(midbody_width, tail_width)
ww[tail:Ns-1]=tail_width
```

In a "reverse skeletonization", we take small rectangular image patches of size $(l, w)$ from the reference image and add them along the target skeleton. Along each skeleton we create

rectangles with $\hat{l}$ oriented along the direction formed by the skeleton points $i$ and $i + step$, and width $w(i) = w_{multiplier} \times ww(i)$. The parameter *step* determines the length $l$ of the rectangle. For each pair of rectangles, we find the affine transformation that maps a rectangle in the reference image to a rectangle in the target image using the function *getAffineTransform* from OpenCV [23]. If *step* is too small (equal to 1), the patches will not overlap which will create discontinuities in the synthetic image, but if *step* is too large, then the patches could be larger than the amount of curvature of the worm. In practice, we set $step = 1/16 \times N_S$. We set $w_{multiplier} = 1.2$, which means the rectangle width will be larger than the actual worm width to include background pixels around the worm body.

For each pair of source-target rectangles, we use the function *warpAffine* from OpenCV [23] to project the pixels from the rectangle in the source image to the coordinates of the target rectangle in the target image. We combine the transformed patches into a single cohesive worm image by iteratively updating a mask image created from the overlapping regions. For each transform, we add the values of the new transformed image containing one patch to the current full image. We then multiply by the mask image set to 1 for non-overlapping areas and 0.5 for overlapping areas. We draw the rectangles from the worm tail so that the last rectangles will be of the worm head, as this configuration is more likely to occur naturally.

The overlapping areas combine seamlessly because of the blending, but some protrusions are still visible, especially when the target pose is very coiled. We eliminate these artifacts by masking the image with a generated image representing the expected worm outline. This mask image is created by drawing convex polygons along the target centerline of the desired worm width, complete with filled circles at the extremities. We apply a median filter with a window size of 3 to smooth the remaining noise due to the joining of the patches. Finally, all non-worm pixels are set to a uniform color: the average of the background pixels in the reference image.

To add diversity to the synthetic images, we include a set of optional augmentations. We translate the target skeleton coordinates by a uniform value between 0 and 5% of the image size. We vary the worm length uniformly between 90% and 110%, and the worm thickness multiplier between 1.1 and 1.3. We randomly switch the drawing order from head to tail or the contrary, so that each is equally probable. Finally, we add an extra Gaussian blur filter at the end of the process 25% of the time, with a blur kernel varying between 3% and 10% of the image size or 13 pixels, whichever is smaller.

In WormPose, the Python implementation of the image generator is optimized for speed and memory allocation. Generating a synthetic image of a large size will be slower than a smaller one. It is also faster to limit the number of reference images, as some calculation is cached. The generation is usually split into several processes, and we use a maximum of 1000 reference images per process, chosen randomly. The number of skeleton points $N_S$ from the reference image is flexible and depends on the dataset. If $N_S$ is too small ($N_S \lesssim 20$), the synthetic worm image will be too simplistic compared to the real images. On the other end, increasing $N_S$ too much will decrease the performance, and the resulting synthetic image will not benefit in detail. We routinely use $50 \lesssim N_S \lesssim 100$.

## Network architecture and training

For reasons ranging from motion blur to self-obscured postures, it is often difficult to discern the worm's head from the tail, such as in Fig 3A. Images with similar worm shape but opposite head-tail locations have quantitatively different centerlines, thus providing a challenge to network training. To handle this ambiguity, we design a loss function that minimizes the difference between the network prediction $\hat{\theta}$ and the closest of two labels: $\theta_a$ and $\theta_b = flip(\theta_a) + \pi$,

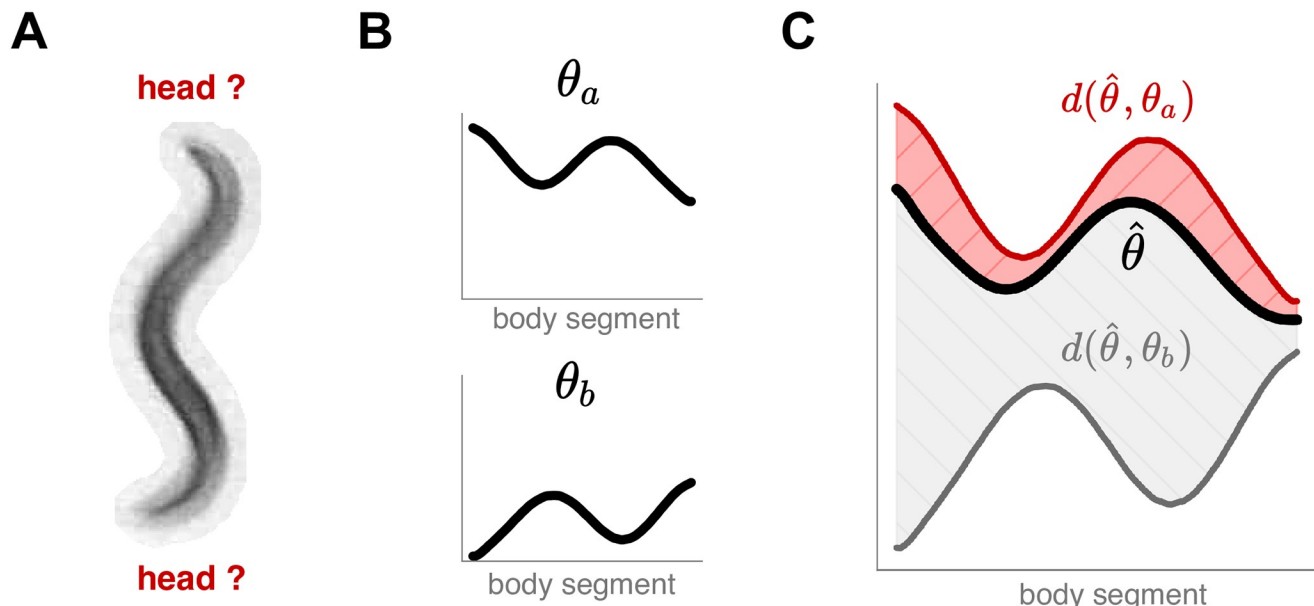

**Fig 3. We train a convolutional network to associate worm images with an unoriented centerline to overcome head-tail ambiguities which are common due to worm behaviors and imaging environments.** (A) An example image with a seemingly symmetrical worm body. (B) We associate each training image to *two* possible centerline geometries, resulting in two equivalent labels: $\theta_a$ and $\theta_b = flip(\theta_a) + \pi$, corresponding to a reversed head/tail orientation. (C) We compare the output centerline $\hat{\theta}$ to each training centerline through the root mean squared error of the angle difference $d(\theta_1, \theta_2)$ (Eq 2) and assign the overall error as $loss = min(d(\hat{\theta}, \theta_a), d(\hat{\theta}, \theta_b))$.

representing the same overall pose but with swapped locations of the head and tail, Fig 3B. The output training error is the minimum of the root mean square error of the angle difference $d(\theta_1, \theta_2)$ between the output centerline $\hat{\theta}$ and the two training labels $\{\theta_a, \theta_b\}$ (Fig 3C) with

$$\epsilon(\alpha, \beta) = atan2(sin(\alpha - \beta), \cos(\alpha - \beta))$$

$$d(\theta_1, \theta_2) = \sqrt{\frac{1}{N} \sum_{i=1}^{N} \epsilon(\theta_{1i}, \theta_{2i})^2} \qquad (2)$$

The learned function is therefore a mapping between the input image and a worm pose without regard to head-tail location, which we determine later with the aid of temporal information.

Our lightweight neural network architecture is heavily inspired by the Residual Network [26], as applied to the CIFAR-10 dataset. Our worm images are bigger than the CIFAR-10 $32 \times 32$ pixels: we routinely pick a linear dimension of 128 pixels, and below 90 pixels the posture become difficult to see. So the first layer of our network is a $7 \times 7$ pixels convolution layer with 32 filters and a stride of 2 pixels, followed by a max-pooling layer with a pool size of $2 \times 2$ pixels and a stride of 2 pixels, which reduces the input image size early in the network. We then use a stack of 3 residual blocks composed of 3 basic blocks, with number of filters: 32, 64, 128. We follow the ResnetV2 architecture [27] where a batch normalization and an activation layer precede the convolution layers. We choose LeakyRelu as the activation layer. The last layers are a global average pooling followed by a densely connected layer with a size of 100.

For each dataset, we generate 500k synthetic images for training, and randomly select 10k real preprocessed images for evaluation. When training, we use a batch size of 128. We train

for 100 epochs and save the model with the smallest error on the evaluation set. We use the Adam optimizer [28] with a learning rate of 0.001.

## Post-prediction

**Image error and outlier detection.**   For real data, the lack of labeled data for coiled worm images means that we cannot directly evaluate the accuracy of the network predictions. Instead, we leverage our ability to generate synthetic images and apply an image error measure between the input image and the two synthetic images generated from the two possible predicted centerlines. We generate synthetic worm images representing the two predictions, using the nearest labeled frame in time as a reference image. We crop the synthetic images to the bounding box of the synthetic worm shape plus a padding of 2 pixels on each side, and apply a template matching function between this synthetic image representing the prediction and the original image. We use the *matchTemplate* function from OpenCV [23] with the normalized correlation coefficient method, which translates a template image across a source image and compute the normalized correlation $c$ at each location. The result is a correlation map, of a size $Size(source) - Size(template) + 1$, with values ranging between $c = -1$ (perfect anti-correlation, as would occur in a pair of reversed-intensity black and white images) and $c = 1$ (perfect correlation). We use the maximum value $|c|_{max}$ to define the image error $1 - |c|_{max}$ and the location of $|c|_{max}$ to estimate the predicted skeleton coordinates. Frames with an image error above a threshold value will be discarded.

To select the threshold (potentially different for each different dataset), we plot the image error distribution on a selection of labeled frames. Comparing images with their reconstructed synthetic image based on their (trusted) labels shows a distribution of low error values, S2 Fig. We select an image error threshold with a default value 0.3, which retains the majority of the predictions while removing obviously incorrect reconstructions.

**Head-tail assignment.**   Once the network is trained, we can predict the centerline in full video sequences, but the resulting postures have a random head-tail assignment. For each image, we augment the predicted centerline $\hat{\theta}$ with the head-tail switched centerline $\hat{\theta}_{flipped} = flip(\hat{\theta}) + \pi$. We use temporal information and the labeled frames to determine the final worm pose as either one of these two centerlines, or we discard the frame entirely in low-confidence cases.

We first create segments with near-continuous poses by using an angle distance function between adjacent frames, $distance(\theta_1, \theta_2) = \frac{1}{N} \sum_{n=0}^{N-1} |\epsilon(\theta_{1n}, \theta_{2n})|$, with $\epsilon$ from Eq 2. We start with the first frame and assign its head position randomly. We then calculate the angle distance between this centerline and the two possible options in the next frame. If the distance is higher than a threshold (we use 30˚), we cannot reliably assign the head position by comparing to this adjacent frame. We calculate the distance on the following frames (maximum 0.2 *s* in the future) until we cannot find any frame that is close enough to the last aligned frame, we then start a new time segment with a random head-tail orientation. After this first process, we obtain temporal segments with a consistent head-tail position, possibly with small gaps containing outlier results to be discarded. To increase confidence in the results, we discard segments that are too small (less than 0.2 seconds).

While the pose of the worm is consistent in these segments, there are still two possible head-tail orientations per segment: we use the labeled data from the non-coiled frames to pick the correct solution. We align the whole segments with the labeled data by calculating a cosine similarity between the head to tail vector coordinates of the prediction and the available labels. We finally align the remaining unaligned segments with no labels by comparing them to the

neighbor segments that have been aligned before: we also calculate the cosine similarity between the head to tail vector between the two closest frames of the aligned and unaligned segment.

**Interpolation.**    For an optional post-prediction step, we interpolate small gaps (max_gap = 4 frames), using a third-order spline interpolation method, using the scipy.interpolate.interp1d function from Scipy [29]].

**Smoothing.**    For an optional post-prediction step, we smooth the angle time series using a Savitsky-Golay filter with third-order polynomials in 8 frame windows, using the scipy.signal.savgol_filter function from Scipy [29].

## Implementation

In Fig 4 we show a schematic of the full computational process, which we implement in a Python package "WormPose", with source code: https://github.com/iteal/wormpose, and documentation: https://iteal.github.io/wormpose/. We optimize for speed via intensive use of multiprocessing and also for big video files that do not fit into memory. We provide default dataset loaders: for the Tierpsy tracker [21] and for a simple folder of images. Users can add their own dataset loader by implementing a simple API: FramesDataset reads the images of the dataset into memory, FeaturesDataset contains the worm features for the labeled frames, and FramePreprocessing contains the image processing logic to segment the worm in images and to calculate the average value of the background pixels. A custom dataset loader is typically a Python module exposing these three objects, which then can be loaded into Worm-Pose by the use of Python entry points. A simplified example of adding a custom dataset is available in the source code repository: https://github.com/iteal/wormpose/tree/master/examples/toy_dataset. We provide a tutorial notebook with sample data and an associated trained model, which can be tested in Google Colaboratory. We also include an optional interface to export results in a custom format: for the Tierpsy tracker dataset, we can export the results to the Worm tracker Commons Object Notation (WCON) format.

We also tested WormPose on a high-performance laptop with a Intel i7-10875H CPU and a NVIDIA GeForce RTX 2070 SUPER GPU, on the same N2 dataset and parameters of the paper. The dataset generation and training the network took 6.5 hours, and predicting the full dataset of 600k+ frames took less than 2 hours. We see no barrier to performing pre-analysis, such as with Tierpsy, on a laptop, thus making the full pipeline accessible.

## Roaming/dwelling analysis

To connect to previous analysis on roaming/dwelling behavior, we compute the worm's speed and angular speed from the centroid $\vec{c} = (x, y)$ position as a function of time. To simplify the comparison with [30], we downsample the time series to 3 Hz and compute the centroid velocity as the finite difference between subsequent time points $\vec{v}(t) = \frac{\vec{c}(t + \Delta t) - \vec{c}(t)}{\Delta t}$, where $\Delta t = 1/3$ s after downsampling. The speed is obtained by taking the norm of the velocity vector $s(t) = |\vec{v}(t)|$, where $|.|$ represents the 2-norm. The angular speed is computed by estimating the angle between the two vectors defined from three subsequent points, which gives the change in the tangential component of the velocity. From these estimates, we obtain roaming and dwelling states by fitting a two-state Hidden Markov Model (HMM) to the speed and angular speed time series averaged in 10 s windows (as in [30]). The model is composed of two hidden states, their stationary distributions $\pi$ and Markov transition matrices $P$, and Gaussian emission probabilities conditioned on the current state. Fitting is performed through an Expectation-Maximization algorithm (Baum-Welch), with the emission probabilities being Gaussian distributions with a diagonal covariance matrix. The sequence of hidden states is

## 0. Data

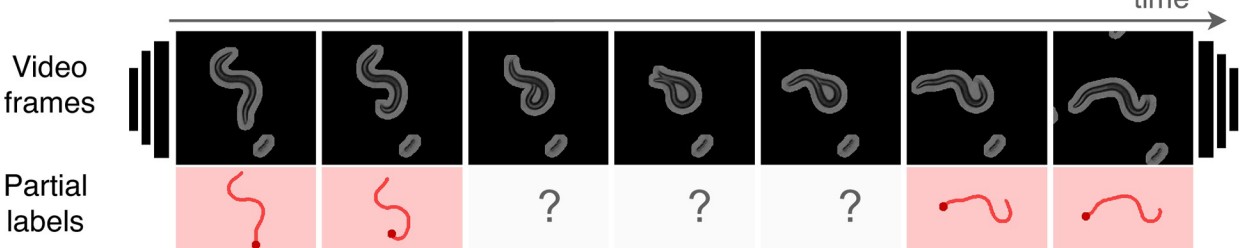

## 1. Train

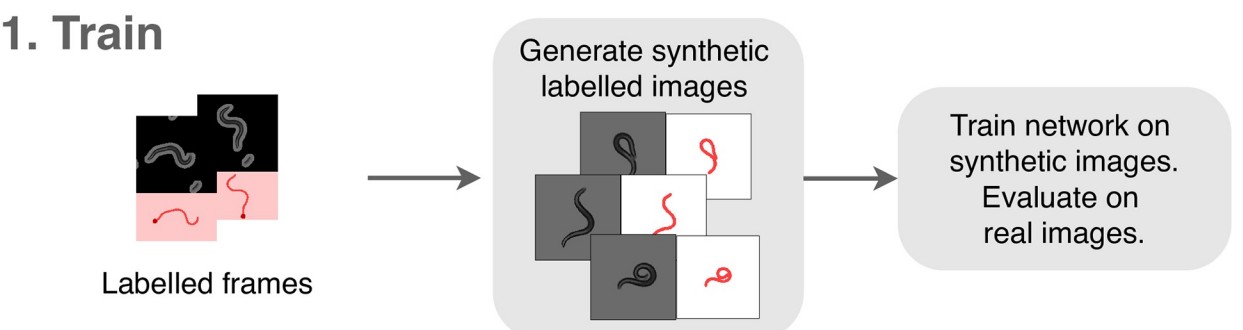

## 2. Predict

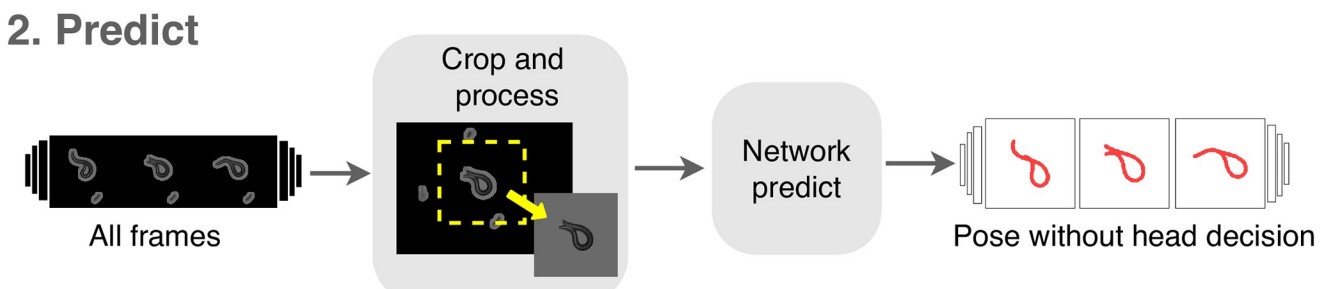

## 3. Process

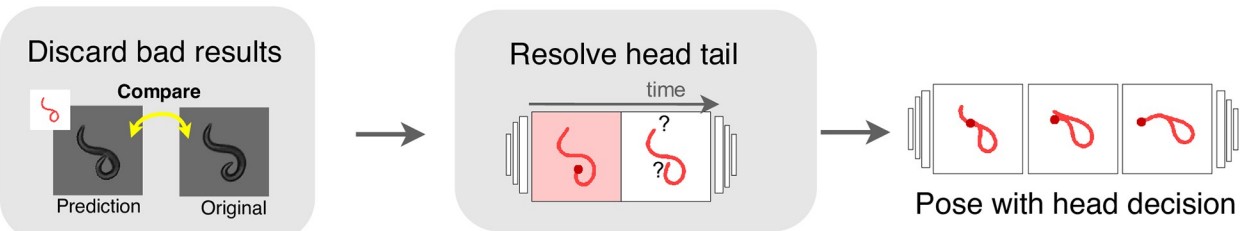

**Fig 4. The WormPose pipeline.** (0) We use classical image processing methods to extract partial labels of simple, non-coiled postures, and then apply a CNN-based approach to complete the missing frames which result from complex images. We analyze each video recording with a three-step pipeline. (1) We generate synthetic data with the visual appearance of the target images but containing a wider range of postures, Fig 2. We use this synthetic data to train a deep neural network to produce the centerline angles from a single image. During training, we periodically evaluate the network on real labeled images and keep the model that best generalizes. (2) We predict the entire set of target images. The images are first cropped and processed to look more visually similar to the synthetic images: background and any non-worm pixels are set to a uniform color. For each such processed image, the trained network predicts the centerline angles for both possible head-tail orientations. (3) Our algorithm produces a full image as output and we discard inaccurate results using a pixel-based comparison with the input image. Finally, we resolve the head-tail orientation by comparing adjacent frames. Once trained, the WormPose pipeline is rapid and robust across videos from a wide variety of recording conditions.

obtained through a Viterbi algorithm. We use an open-source Python HMM package, `hmmlearn`, obtained from: https://github.com/hmmlearn/hmmlearn. For more on HMMs, see [31].

To determine the directionality of the worms movement, we estimate the tangential component of the velocity vector by $\psi(t) = \tan^{-1}(v_y(t)/v_x(t))$, and the overall tail-to-head angle by averaging the centerline angles $\Psi(t) = \langle \vec{\theta}(t) \rangle$. The worms orientation at each time point is obtained by subtracting these two quantities $\Delta\psi = \psi - \Psi$ and normalizing into the interval $\left[ -\frac{\pi}{2}, \frac{3\pi}{2} \right]$ [32].

Posture analysis was performed by projecting the centerline angle time series $\vec{\theta}(t)$ into a canonical lower dimensional space of "eigenworms" [7], resulting in a mode time series $\vec{a}(t)$. In this space, the first two eigenmodes capture the propagation of the body wave along the body. The angle between them, $\phi(t) = \tan^{-1}(a_2(t)/a_1(t))$, defines the phase of the wave, while its derivative $\omega(t) = \dot{\phi}(t)$ is the phase velocity. Estimates of the phase velocity $\omega$ are obtained through fitting a cubic spline to $\phi$, using Scipy's `interpolate.CubicSpline` [29]. We estimated the frequency of complete body waves by finding segments in which the body wave phase velocity $\omega$ did not change sign, and there is a recurrence in $\cos(\phi(t))$. We make a conservative estimate of the body wave frequency by counting peaks in the time series of $\cos(\phi(t))$, using the `scipy.signal.find_peaks` function of Scipy [29], with a prominence 1.95 and a minimum time between peaks of 8 frames ($\sim 0.27$ s).

For comparison to off-food behavior we used a previously-analyzed dataset [9, 33], in which N2-strain *C. elegans* were imaged at $f = 32$ Hz with a video tracking microscope on a food-free plate. Worms were grown at $20°C$ under standard conditions [34]. Before imaging, worms were removed from bacteria-strewn agar plates using a platinum worm pick, and rinsed from *E. coli* by letting them swim for 1 min in NGM buffer. They were then transferred to an assay plate (9 cm Petri dish) that contained a copper ring (5.1 cm inner diameter) pressed into the agar surface, preventing the worm from reaching the side of the plate. Recording started approximately 5 min after the transfer, and lasted for 2100 s. In our analysis we used data downsampled to $f = 16$ Hz [9], yielding 33600 frames per recording.

## Results

### Pose estimation from wild-type and mutant worm recordings

We quantify WormPose using synthetic data as well as (N = 24) wild-type N2 worm recordings and (N = 24) AQ2934 mutants from the Open Worm Movement Database. The synthetic data analyzed here was not used for training and consists of 600k images. We choose N2 for general interest and AQ2934 (with gene mutation nca-2 and nRHO-1) for the prevalence of coiled shapes. For the AQ2934 dataset, we used all of the available videos. For the N2 dataset, we selected 24 videos randomly from the large selection in the Open Worm Movement Database, but with a criterion of a high ratio of successfully analyzed frames from the Tierpsy Tracker (in practice ranging between 79% to 94%). Videos where there are very few analyzed frames may signal that the worm goes out of frame, or that the image quality is so low that no further analysis is possible. Images are sampled at rate $f_s \sim 30$ hz for $\sim 15$ min in duration, resulting in 600k frames from each dataset. We set the image size to $128 \times 128$ pixels. We train distinct models for each dataset and then predict all images from each dataset. We show the cumulative distribution of the image error in Fig 5A, including typical (input and output) worm images for various error values. For additional context, we also show the image error calculated on synthetic image data not used in training. Errors in the synthetic data are larger than those for N2 worms because we have more (and more complicated) coiled postures in the

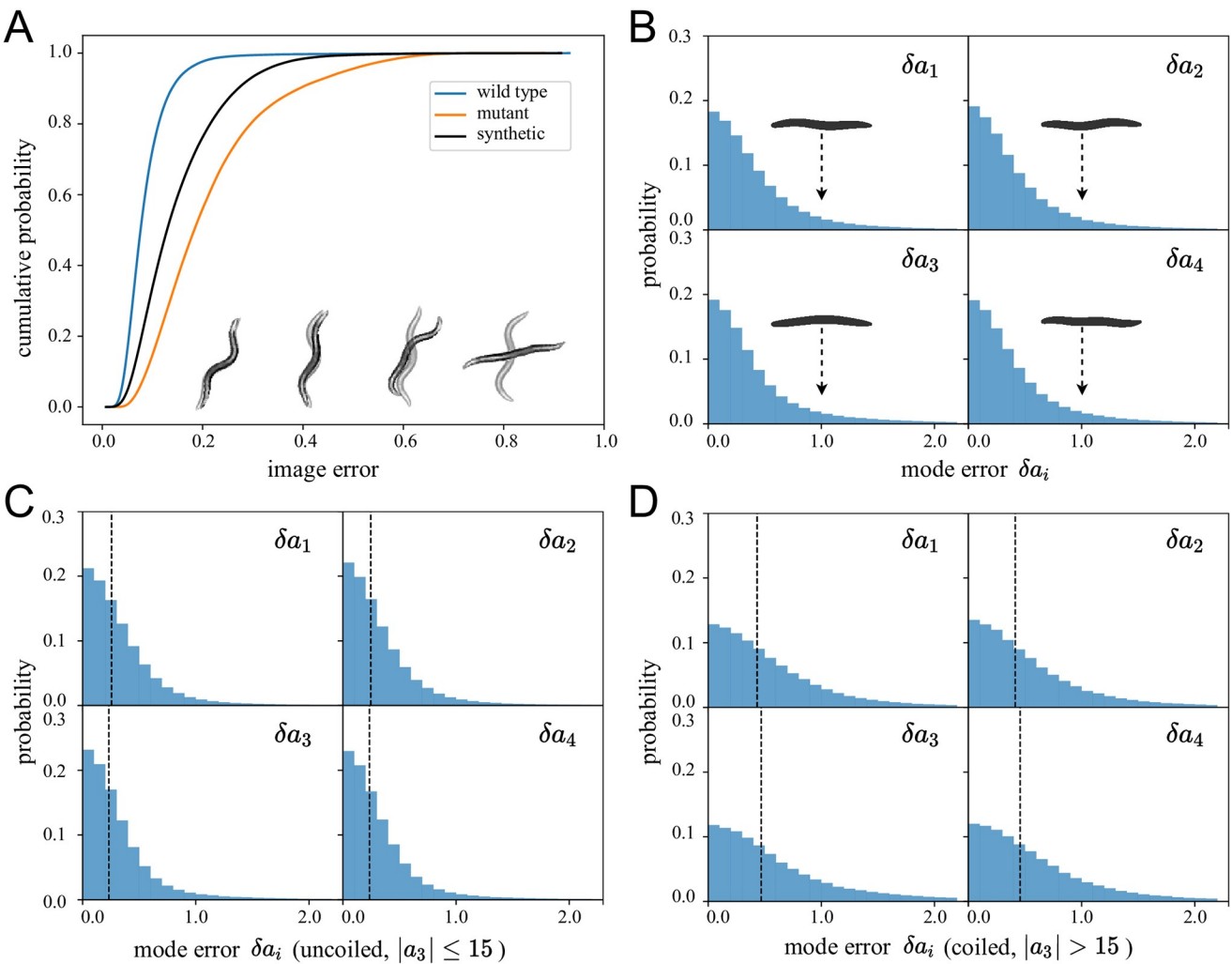

**Fig 5. Quantifying the error in pose estimation.** (A) We show the cumulative image error of predicted images for different datasets. We predict 24 videos totaling over 600k frames from N2 wild-type and AQ2934 mutant datasets and calculate the image error between the original image and the two possible predictions, and keep the lowest value between the two. For the error calculations here we bypass the postprocessing step so no result is discarded. For interpretability we also draw representative worm image pairs for different error values and note that predictions overwhelmingly result in barely discernible image errors. On average, the N2 predictions have a lower image error than the mutant which exhibit much more coiled challenging postures. We also generate new synthetic images (using N2 as templates, 600k values) not seen during the training and predict them in the same way. The image error for the synthetic images (which generally include a higher fraction of complex, coiled shapes) is on average worse than the N2 type, but better than the mutant. (B) Our synthetic training approach also allows for a direct comparison between input and output centerlines, here quantified through the difference in eigenworm mode values. As with the images, the differences are also small so that even in the large-error tail of the distribution the "error worms" (worm shapes representing mode values with $\delta a_i = 1.0$) are essentially flat. The median mode errors are $\langle \delta \vec{a} \rangle_{\text{median}} = (0.30, 0.29, 0.29, 0.29)$. (C, D) We additionally show the mode errors for synthetic images separated into (C) uncoiled ($a_3 \leq 15$) and (D) coiled ($a_3 > 15$) shapes. Dashed lines denote median error values of $\langle \delta \vec{a} \rangle_{\text{uncoiled}} = (0.26, 0.25, 0.23, 0.23)$ and $\langle \delta \vec{a} \rangle_{\text{coiled}} = (0.43, 0.41, 0.47, 0.46)$. The errors are small in all cases.

synthetic data generator. In Fig 5B we use our image generator to show the error in mode values for synthetic data, the only data for which we have ground truth for the centerlines. The "error worms" (worm shapes representing mode values with $\delta a_i = 1.0$) are essentially flat and we report even smaller median mode errors $\langle \delta \vec{a} \rangle_{\text{median}} = (0.30, 0.29, 0.29, 0.29)$. In Fig 5C and 5D we show that the errors remain small when separated into uncoiled and coiled shapes (see [9] for a comparison).

## Comparison with previous approaches

The only comparable open-source, coiled-shape solution is detailed in previous work from some of the current authors [9] (hereafter noted as RCS from an abbreviation of the title). RCS was designed before the widespread application of CNNs and was evaluated entirely on postures from N2 worms. For coiled frames, RCS employs a computationally expensive pattern search in the space of binarized down-scaled worm images, thus ignoring texture and other greyscale information. A temporal algorithm then matches several solutions across frames to resolve ambiguities. We apply RCS to the N2 and mutant AQ2934 datasets analyzed above. The mutant dataset is especially challenging as a large proportion of coiled frames require the slow pattern search algorithm. We split each video into segments of approximately 500 frames to parallelize the computation on the OIST HPC cluster and obtain results in approximately one week while running 100 cluster jobs simultaneously. For comparison, WormPose applied to the mutant data completed in approximately a day while running only one job on a GPU node with an Nvidia Tesla V100 16GB, with the majority of the time spent on network training. Ultimately we obtained posture estimates for 98% of the frames of the mutant dataset and 99.8% of the N2 dataset.

Unfortunately, a lack of ground truth posture sequences means that we cannot directly compare the posture estimates of RCS and WormPose. Posture sequences are fundamental to RCS and this information is not contained in the image generator of WormPose. However, we can leverage the image error between the original image and the predicted posture (without head information), S3 Fig. While WormPose is dramatically faster and uses no temporal information (a possible route for future improvement), we obtain very similar image reconstruction errors for both methods (A). For a closer examination, we also show cumulative distributions of the difference in turning mode values (B). One source of these discrepancies are coiled loop-like postures where both methods struggle to recover the correct pose. Another discrepancy (C) results from crossings such as illustrated in Fig 1 where RCS's temporal matching algorithm picks the wrong solution, perhaps a reflection of loss of information upon binarization.

## Posture-scale analysis of roaming/dwelling behavior

We further demonstrate WormPose by exploring previously unanalyzed $N = 8$ longtime ($T \sim 8$ h, $f_s \sim 30$ hz) recordings of on-food N2 worms. The length of these recordings ($\mathcal{O}(10^6)$ frames) renders impractical previous coiled shape solutions [9, 10], which has prevented fine-scale posture analysis of roaming/dwelling behavior.

On food-rich environments, worms typically switch between two long-lasting behaviors: a roaming state, in which worms move abundantly on the plate at higher speeds and relatively straight paths; and a dwelling state, in which worms stay on a local patch with lower speeds and higher angular speeds [30, 35]. Roaming and dwelling states can last for tens of minutes, so long recordings are essential, and we leverage our ability to obtain high-resolution posture tracking to explore their fine-scale behavioral details.

To identify roaming and dwelling states consistent with previous work [30], we fit a Hidden Markov Model to the centroid and angular speed averaged in 10 s windows, which yields a high speed, low angular speed state (roaming) and a low speed, high angular speed state (dwelling), Fig 6A. We estimate the frame-by-frame directionality of the worm's movement by subtracting the angle of the velocity vector $\psi = \tan^{-1}\left(\frac{v_y}{v_x}\right)$ (where $v_x$ and $v_y$ are the $x$ and $y$ components of the centroid velocity) by the overall tail-to-head worm angle on the plate $\Psi$, obtained by averaging the centerline angle along the body, $\Delta\psi = \psi - \Psi$. The distribution of $\Delta\psi$

 WormPose: Image synthesis and convolutional networks for pose estimation in *C. elegans*

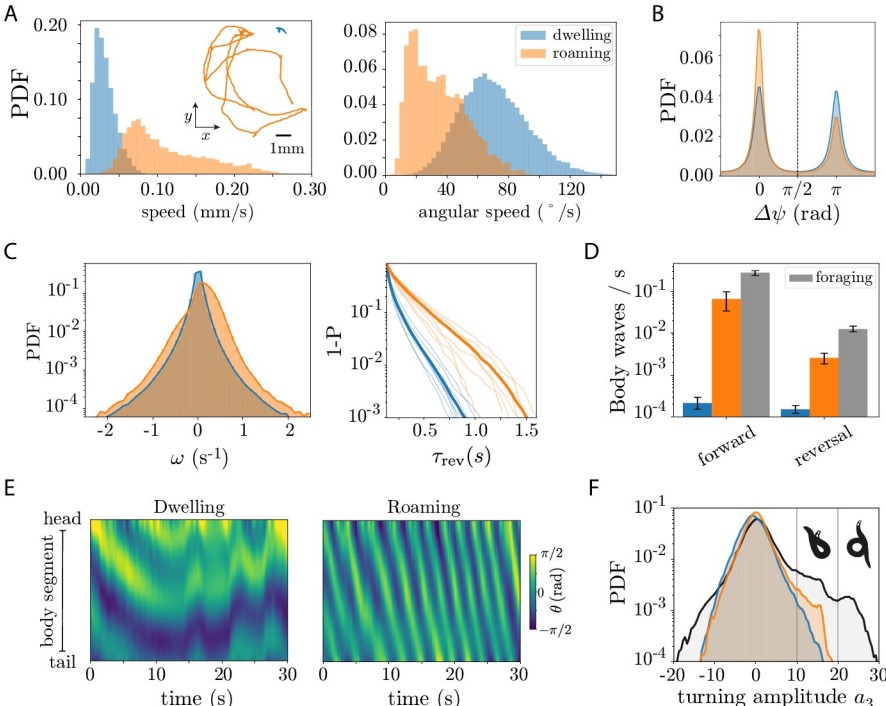

**Fig 6. Posture-scale analysis of roaming/dwelling behavior from long ($T \sim 8\,h$) recordings reveals that the centroid-derived increase in the dwelling reversal rate results from incoherent body motions that do not translate the worm, and that deep ventral turns are less common in on-food vs off-food conditions.** (A) We align with previous definitions by identifying roaming/dwelling behavior through a Hidden Markov Model of the linear and angular speed, averaged in 10 s windows, thus splitting each trajectory into two states: a low speed, high angular speed state (dwelling, blue), and a high speed, low angular speed state (roaming, orange). (inset) Example 5 minute centroid trajectories for each state. (B) In centroid-based analysis, the dwelling state exhibits a larger fraction of reversals vs roaming. We identify forward and backward motion using the angle between the centroid velocity vector and the tail-to-head angle obtained by averaging centerline angles: $\Delta\psi < \pi/2$ for forward locomotion, $\Delta\psi > \pi/2$ for backwards. (C-F) Posture analysis reveals that the centroid characterisation of roaming and dwelling behavior is incomplete. (C, left) Roaming worms exhibit a larger fraction of higher body wave phase velocities, $\omega$, in both reversal and forward motion. (C, right) Probability of reversals longer that $\tau_{\rm rev}$, $P(t > \tau_{\rm rev}) = 1 - P(t \leq \tau_{\rm rev})$ in the dwelling and roaming states. The roaming state generally exhibits longer reversals than dwelling, for which reversal bouts are extremely short. Thick lines indicate the CDF for the ensemble of worms, while lighter lines are for each individual. (D) The rate of reversal events with complete body waves is an order of magnitude higher in the roaming state compared to dwelling. For comparison we show also the reversal rate for worms foraging off-food [7, 9] (gray), which also exhibits an increased reversal rate. For such body wave analysis we identify forward and reversal events according to the sign of the phase velocity $\omega$. (E) Body curvature $\theta$ as a function of time for example dwelling and roaming states. The dwelling state (left) exhibits incoherent body waves that do propagate through the entire body, whereas coherent full body waves are commonly observed in roaming (right). (F) Worms on-food exhibit a lower fraction of deep ventral turns. We show the probability distribution function (PDF) of the turning mode $a_3$ for roaming, dwelling and foraging (gray) worms. Roaming worms exhibit a larger fraction of $\Omega$-turns than dwelling worms and $\delta$-turns are rare in on-food data: they are not observed in the full $\sim$ 66 hours of recordings.

is bimodal, indicative of switching between forward ($\Delta\psi \approx 0$ rad) and reversal ($\Delta\psi \approx \pi$ rad) movement, Fig 6B. As in previous observations [30, 35], worms mostly move forward in the roaming state, while dwelling exhibits a larger fraction of backward locomotion.

Our high-resolution posture measurements provide a unique opportunity to dissect the fine-scale details of these long time scale behaviors; WormPose allows us to substantially reduce noise in the estimate of the body wave phase velocity resulting from blurry frames in the long recordings, and to obtain the full body posture through coiling events. We leverage the interpretability of the eigenworm decomposition of the centerline angles [7] to assess the properties of the body wave. The first two eigenworms ($a_1$ and $a_2$) capture the undulatory

motion of the worm: the angle between these two modes $\phi = -\tan^{-1}\left(\frac{a_2}{a_1}\right)$ is the overall phase of the body wave, while its derivative, $\dot{\phi} = \omega$ is the body wave phase velocity. The third eigen-worm, $a_3$, captures the overall turning amplitude of the worm: $|a_3| \gtrsim 10$ correspond to $\Omega$-like turns [7, 9]. In Fig 6C we show the distribution of phase velocities $\omega$, in the roaming and dwelling states previously identified. Roaming worms typically exhibit higher body wave phase velocities in both forward ($\omega > 0$) and backward ($\omega < 0$) locomotion, Fig 6C(left), contrary to the centroid characterization, which indicates that dwelling worms increase their rate of reversals and reorientation events [30, 35]. Notably, most reversals in the dwelling state are very short (90% of them are shorter than $\sim 0.25$ s), when compared to the typical reversal length in the roaming state, Fig 6C(right). The low phase velocities in dwelling also indicate that such reversals result in an insignificant translation of the worm's body. This suggests that most of the reversals measured through a centroid-based analysis in fact correspond to incoherent body motions, such as head oscillations or short retractions. Indeed, we count the frequency of body waves that travel all the way across the body, Fig 6D, and find that the frequency of full-body waves is extremely small in dwelling when compared to the roaming, for which coherent body movements are much more frequent. Comparison to off-food behavior indicates that foraging worms exhibit an even higher rate of full body waves in the forward and reversal state, contradicting the centroid-derived picture that an increase rate of reversals results in local exploration.

While dwelling states at the centroid level exhibit larger reversal rates, the nature of these reversals is very different from the coherent body wave reversals found during roaming. In Fig 6E we show examples of 30s segments of the body angles as a function of time, illustrating how apparent reversals in the dwelling state result from incoherent body motions. To further dissect the nature of roaming and dwelling states, we leverage WormPose to compute the distribution of turning amplitudes $a_3$ across states, Fig 6F. Roaming states exhibit a slightly larger fraction of $\Omega$-turns ($10 \lesssim a_3 < 20$) when compared to dwelling worms, which contradicts the centroid-derived picture (prevalent in previous literature) that dwelling results from increased reversal and turning rates. Remarkably, in a total of $\sim 66$ hours of analyzed data in on-food conditions, we find no occurrence of deep $\delta$-turns, a behavior commonly observed when foraging off food.

## Availability & future directions

WormPose is open-source and free with a permissive 3-Clause BSD License. The source code is available: https://github.com/iteal/wormpose, and can be installed from the Python package index: https://pypi.org/project/wormpose. The GitHub README also includes a link to the data used in our analyses. The GitHub material includes scripts to download on-food datasets from Zenodo, as well as trained models.

## Discussion

WormPose enables 2D pose estimation of *C. elegans* by combining a CNN with a synthetic worm image generator for training without manually labeled data. Our approach is especially applicable to complex, coiled shapes, which have received less attention in quantitative analyses even as they occur during important turning behaviors and in a variety of mutants. We also introduce an image similarity, which leverages the synthetic worm generator to assess the quality of the predicted pose without manual centerline annotation. Once trained, the convolution computation is fast and could enable real-time, coiled-pose estimation and feedback

[36]. The computational pipeline is optimized to analyze large datasets efficiently and is packaged in an easy to use, install and extend, open-source Python package.

With common imaging resolutions, the determination of the worm's head-tail orientation is surprisingly subtle. Our approach uses the presence of labeled trusted frames from traditional tracking methods which rely on brightness changes or velocity. An appealing alternative would be to estimate the head location directly. For example, [37] uses a network to regress the coordinates of *C. elegans* head and tail. In addition, CNN's that estimate keypoint positions [16], [17], [18] are now widely available. However, such current general techniques applied to ambiguous worm images result in low-confidence head-tail location probabilities, especially for blurry, low-resolution or self-occluded images. Training for this task is noisy and slow to converge, suggesting that there is simply not enough visual information in a single image.

Our posture model necessitates a library of examples which we obtained from N2 worms. Some strains however have different postures such as *lon-2* or *dpy* which are longer and shorter than N2, respectively. In particular, *lon-2* can make more coils due to its longer body, and our posture model does not represent the wider variety of possible postures. Of course, we can always augment the posture library. But a more general solution is to create a physical model of the worm [38].

Our approach follows advances in human eye gaze and hand pose estimation where it is difficult to obtain accurate labeled data. 3D Computer Graphics are often employed to create synthetic images [39] with increasing realism [40]. Synthetic images for human pose estimation have also been created by combining and blending small images corresponding to the body limbs of a labeled image, to form new realistic images [41]. To bridge the similarity gap between the real and the synthetic domain, Generative Adversarial Networks (GAN) techniques alter such computer-generated images [42] or directly generate synthetic images from a source image and a target pose [43]. Models of the deformable source object (e.g. human limbs) are often encoded into such generative networks to avoid unrealistic results. Some of these ideas have been recently applied to laboratory organisms [44], including *C. elegans*, but have avoided the fundamental complexity of self-occluding shapes. Outside of the laboratory, [45] proposes an end-to-end approach to estimate zebra pose using a synthetic dataset and jointly estimating a model of the animal pose with a texture map. Another approach is to adversarially train a feature discriminator until the features from the synthetic and real domain are indistinguishable [46, 47]. In both humans and animals, we expect that the combination of physical body models and image synthesis will be important for future progress in precise pose estimation.

## Supporting information

**S1 Fig. Model selection assessment in the Gaussian Mixture Model of worm shapes.**
(A) Akaike Information Criterion for GMMs with different numbers of gaussian components. The minimum is attained with $N = 270$ gaussian components. Error bars represent 95% confidence intervals over 100 different training sets of $\sim 15000$ worm shapes sampled uniformly according to the body curvature as measured by the third eigenworm coefficient, $a_3$.
(B) Covariance matrix of the space of mean subtracted tangent angles $\vec{\theta}$ for the data used in training (left) and an equal number of simulated angles (right).
(TIF)

**S2 Fig. The cumulative distribution of the image error for all available labeled (and thus uncoiled) frames in the N2 wild-type and AQ2934 mutant datasets.**
(TIF)

**S3 Fig. Comparing WormPose to a reference method (RCS) [9].** (A) We show the cumulative image error of predicted images, similarly to Fig 5A. While the image error is similar, WormPose is faster and does not make use of temporal information (a possible route for future improvement). (B) Cumulative distributions of the difference in $a_3$ mode values $\delta = \|WP_{a_3}\| - \|RCS_{a_3}\|$, restricted to coiled shapes ($|a_3| > 15$) and image error $\leq 0.3$ as determined from the output of WormPose. We plot separate distributions for the wild-type and mutant strains. Large deviations between the methods occur primarily in the coiled mutants and we manually examine a subset of 100 images with $\delta > 10$ (a difference chosen to facilitate comparisons by eye) where we find 72% correctly tracked by WormPose, 6% correctly tracked by RCS, and 22% in which the better tracked centerline was unclear. A video of this inspection process is available with the data. (C) Qualitative results for a selection of frames where the image error doesn't fully describe the discrepancies between the two methods. Very tight loops (top) are challenging for both methods and RCS typically misidentifies crossings where grey-scale information would help (middle and bottom).
(TIF)

## Acknowledgments

We thank Mathijs Rozemuller (AMOLF) for code testing and for providing a tutorial dataset, as well as Jarlath Rodgers (University of Toronto) and Kelimar Diaz Cruz (Georgia Tech) for code testing. We are also grateful for the help and support provided by the Scientific Computing section of Research Support Division at OIST.

## Author Contributions

**Conceptualization:** Laetitia Hebert, Greg J. Stephens.

**Data curation:** Laetitia Hebert, Antonio C. Costa, Liam O'Shaughnessy, Greg J. Stephens.

**Formal analysis:** Laetitia Hebert, Antonio C. Costa, Liam O'Shaughnessy, Greg J. Stephens.

**Funding acquisition:** Greg J. Stephens.

**Investigation:** Laetitia Hebert, Tosif Ahamed, Antonio C. Costa, Liam O'Shaughnessy, Greg J. Stephens.

**Methodology:** Laetitia Hebert, Antonio C. Costa, Greg J. Stephens.

**Project administration:** Laetitia Hebert, Greg J. Stephens.

**Resources:** Laetitia Hebert, Greg J. Stephens.

**Software:** Laetitia Hebert, Antonio C. Costa.

**Supervision:** Greg J. Stephens.

**Validation:** Laetitia Hebert, Tosif Ahamed, Antonio C. Costa, Liam O'Shaughnessy, Greg J. Stephens.

**Visualization:** Laetitia Hebert, Antonio C. Costa, Liam O'Shaughnessy, Greg J. Stephens.

**Writing – original draft:** Laetitia Hebert, Tosif Ahamed, Antonio C. Costa, Liam O'Shaughnessy, Greg J. Stephens.

**Writing – review & editing:** Laetitia Hebert, Tosif Ahamed, Antonio C. Costa, Liam O'Shaughnessy, Greg J. Stephens.

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
