## [Decision Letter · Decision Letter 0]

12 Dec 2020

Dear Prof. Stephens,

Thank you very much for submitting your manuscript "WormPose: Image synthesis and convolutional networks for pose estimation in C. elegans" for consideration at PLOS Computational Biology. As with all papers reviewed by the journal, your manuscript was reviewed by members of the editorial board and by several independent reviewers. The reviewers appreciated the attention to an important topic. Based on the reviews, we are likely to accept this manuscript for publication, providing that you modify the manuscript according to the review recommendations.

Sincerely,

Dina Schneidman

Software Editor

PLOS Computational Biology

[LINK]

Reviewer's Responses to Questions

**Comments to the Authors:**

Reviewer #1: "WormPose: Image synthesis and convolutional networks for pose estimation in C. elegans" by Hebert et al. introduces a new algorithm for resolving coiled postures in the nematode worm C. elegans. Progress in tracking and analysing worm behaviour has been foundational in the study of the physics of behaviour and has contributed to behavioural genetics and phentoypic drug screens. Coiled postures are an important element of C. elegans behaviour because they adopt them for sharp reorientations that are used in taxis and escape. However, because the pose of coiled worms is difficult to estimate, coiled shapes have contributed less to the analysis of worm behaviour than they should. The improved method presented here is thus an important contribution to C. elegans behaviour research.

The method can also be seen in the context of deep learning methods applied to animal behaviour which have seen a recent explosion of interest. Much of this interest has been in the fact that complex animals in complex scenes can finally be tracked at all, but there has been less attention to the accuracy of the tracking. I appreciate the lenghts the authors go to in this paper to quantify the quality of the tracking results.

The final important aspect of the method is that it works without large sets of manually labelled training images which is an advantage compared to most other methods. Even in cases where manually annotated data are available, I can imagine their image synthesis method being used for augmentation.

The paper is well written and clear, the figures are well-presented, and the reported software is accessible and well-documented. In my opinion the paper is publishable as-is. My comments below should be seen only as suggestions for possible improvement.

Specific comments:

-when discussing previous methods of solving the coiling problem Huang et al. (2006) J. Neurosci. Meth. should be cited as well.

-"CNNs are the foundation for recent, remark- able progress in markerless body point tracking [15–17], including worm posture [18, 19]. However, intensive la- beling requirements by human annotators, even if as- sisted by technology [20], as well as the ambiguity of which or exactly how many points to label, offer a bar- rier to the usefulness of CNNs in posture tracking and beyond." If I've understood correctly, the method in reference 18 also works without manually annotated images so the discussion should be modified to reflect that.

-in the network architecture and training section more detail on the CNN should be provided (type of CNN (ResNet), how many layers, neurons per layer, etc.)

-typo: "we set set the weights"

-C. elegans not italicised in second paragraph of discussion.

Reviewer #2: Summary

The authors develop a neural network-based algorithm and open source package in python for reconstructing postures of the model organism C. elegans. Previous work successfully reconstructs postures for simple and complex body shapes, and this work is tailored at reconstructing shapes difficult to resolve. In addition, the computational time of this algorithm is orders of magnitude faster than the most similar related work, making the approach largely feasible now. This algorithm is trained on pixels directly, and includes a generative model that can simulate large quantities of images with a ground truth posture. This step allows small quantities of previously labeled images to be expanded to arbitrary amounts of training data, allowing automation of the entire pipeline. Finally, the authors apply their algorithm to analyze a large posture dataset, producing new insights about the organization of actions within roaming-dwelling two-state switching behavior.

Detailed Review

Overall this manuscript describes an appropriate use of an exciting technology (neural networks) to an outstanding problem of biological interest (posture analysis), and overcomes some key limitations of previous work (extremely long computational time). The main issue that we see is appropriate error quantifications, also with respect to direct comparability with previous work. Thus should be straight-forward to fix. The following is organized into several sections:

1) Various error quantifications

Specifically, the errors reported, e.g. in Fig. 5, are not most relevant to broader usefulness of the algorithm to experimentalists and the interpretable output. We would like to see a quantification of errors but for ground truth centerlines in real data.

The use of pixel error is less interpretable and does not address the difficult problems this algorithm aims to solve. This is particularly important for cases like that shown in Fig. 1C, where the pixel error may be very small, but the posture error may be very large. Indeed, how often does this algorithm correctly distinguish between the two possibilities in Fig. 1C? We wonder whether there could be a direct and intuitive metric used for evaluation, for example the fraction of correctly annotated centerline crossings.

Related, the underlying data are effectively split into easy postures (straight motion) and very difficult postures (coiled and self-occluding). Thus, it seems important to the claims of the paper to characterize the algorithm error separately for these qualitatively different clusters of postures, as in the similar panel of Fig. 2F in related previous work [1].

Similarly, the comparison to previous work is incomplete. Figure S4 compares the pixel-wise errors between this work and a previous algorithm, but given that the scientific output of this algorithm is centerlines, a direct comparison of centerline error seems necessary. In the second results section the authors state “Unfortunately, a lack of ground truth posture sequences means that we cannot directly compare the posture estimates of RCS and WormPose.” However, a smaller set of manual annotations can be generated to produce this comparison.

A different step of the algorithm also appears to require an additional error reporting: the head-tail discrimination module. While the methodology is well explained, the overall fraction of correctly annotated heads/tails is not reported.

2) Minor data notes

The authors state “we also expect that the labeled frames provide the head-tail position at regular intervals throughout the video". Is there a GUI or expected format for this input? Although this requirement leaves the following statement in the abstract technically correct “thus avoiding the need for human-labeled training”, it should be noted more prominently that analyzing a new dataset is not completely automated.

Related, the authors assume “that the input data consists of videos of a single moving worm and that most of the non-coiled frames are analyzed beforehand”, but Section Results/”Comparison with previous approaches” states that WormPose can be performed on a laptop. A sentence about whether the entire pipeline (i.e. including the required pre-analysis) can also be done on a laptop should be included.

3) Minor algorithm notes

The authors use a GMM to build their generative model. It is unclear why this choice was made. We assume this method was chosen because it is simple, fast, common, and generative. However, this should be explained.

In Fig. S2A, the authors use AIC to determine the number of components of their GMM. However, two things are unclear: Why was AIC chosen and not cross-validation? What are the error bars on the plot? Although it does not seem to affect the results at all, a sentence or two of explanation would be appreciated.

4) Minor other notes and typos

The videos on the GitHub page are nice; some similar videos and definitely more of them should be included with the supplementary information.

In the second paragraph of section Methods/”Processing Natural Images” the quotes are wrong in the phrase:

The morphological operation ”close”

The caption of Fig. S1 should be “generate” not “generated”

5) Other computational notes

We commend the authors for a well presented open-source package. In particular, it is excellent that the package is installable via pip and the requirements are included, with versions, in the repository.

We believe that the computational speed of this package is a very strong asset, and could be emphasized more as an important scientific contribution.

References

[1] Broekmans, O.D., Rodgers, J.B., Ryu, W.S. and Stephens, G.J., 2016. Resolving coiled shapes reveals new reorientation behaviors in C. elegans. Elife, 5, p.e17227.

Reviewer #3: Summary:

This paper addresses problems with resolving coiled/self-occluding shapes in worms. Conventional image skeletonization methods work for uncoiled shapes but fail when worms self-intersect. For these coiled shapes, this group has previously implemented a method which resolves centerlines by searching the space of eigenworms for a reasonable match, however this method is time consuming and computationally expensive. Other groups have used convolutional neural networks to track organisms (e.g. DeepLabCut) without labels and have even applied CNNs to tracking worm centerlines. The authors claim that these CNN based methods are limited because they require human annotation as an input, so they have devised a scheme for creating synthetic data based on traditional image skeletonization (uncoiled shapes) and the eigenmode projection method in ref 9 (for coiled shapes) as a training data set for the CNN. This is the primary new contribution in the paper. The paper will certainly be of interest to researchers studying nematode movement, but it would nice to know if the technique is restricted to "roaming and dwelling" behaviors or can be applied to movement in more complex environments (and other organisms).

Here are some broad comments which would be valuable for the authors to address at least in response letter, and could potentially make it into the manuscript to increase biological relevance:

Overall, I think the method is sound and useful, however I wonder about its domain of applicability. Since it's based on eigenmode projections, its not clear to me that it will work efficiently in cases where worms are no longer well captured by the eigenworms derived from observations of agar crawling. This might include worms in more complex environments. If they could show that, for instance, thrashing worms could still be reasonably well resolved, or mutants that display radically different mechanical properties, rolling mutants etc. could be well captured with this eigenworm driven training set that would be valuable. I'm not sure if that would just reflect that robustness of the CNN or if that would suggest that the eigenworms from agar are generic enough to be pushed into new territory. In terms of domain of application, what about other problems where linear shapes are bent into self-occluding shapes (e.g. snakes or plant roots)?

If the use is limited to worms on agar, that's still a pretty wide community and essentially the tool is good for identifying nuances of turning behavior for worms on agar at scale. This suggests a question, why not just use the centroid information to perform behavioral phenotyping and studying long time behavior? Or perhaps centroid + fitting the worm to an ellipsoid to get overall angle information? What do you gain by having postural information at this scale? They attempt to address this in the last section of the results (posture-scale analysis of roaming/dwelling behavior).

It is certainly impressive and a little tantalizing to have posture-level resolution in a 10 hour, 30 Hz experiment, however, I think they could have done a little more digging to say what is gained with this extremely fine grained analysis. They assess some subtleties of the difference between roaming and dwelling states which have previously been identified by simply looking at raw motion, however I don't know how salient these subtleties are, moreover the basic identification of these behavioral states is not upended by the details. Can the postural analysis make the definition of these states more robust, rather than just using centroid information to define the states and then commenting on subtle postural differences in the centroid-determined states? The most striking detail they uncover is in Fig. 6 E, but this doesn't really engage the self-occluded shapes, since these are forward and reverse travelling body waves? I don't dispute that these details are interesting, but they could be pushed further, though that may be beyond the scope of this paper.

My last question is this, in previous work (ref 9) resolution of self-occluded shapes revealed a distinction between the delta turn and the omega turn. Since this technique allows a dramatic scale up in the number of resolvable self-intersecting states, does the larger data set shed any light on novel details of turning? Can turning behaviors be even more subtly delineated with the higher statistical resolution the technique allows?

Here are some specific (nitpicking) comments: I would like to see terms like "generative" defined. And I would hardly call the images presented in the manuscript "Natural" images!

**Have all data underlying the figures and results presented in the manuscript been provided?**

Reviewer #1: Yes

Reviewer #2: Yes

Reviewer #3: Yes

PLOS authors have the option to publish the peer review history of their article (what does this mean?). If published, this will include your full peer review and any attached files.

Reviewer #1: No

Reviewer #2: No

Reviewer #3: No
---

## [Editor Report · Decision Letter 1]

25 Mar 2021

Dear Prof. Stephens,

We are pleased to inform you that your manuscript 'WormPose: Image synthesis and convolutional networks for pose estimation in C. elegans' has been provisionally accepted for publication in PLOS Computational Biology.

Best regards,

Dina Schneidman

Software Editor

PLOS Computational Biology

---

## [Editor Report · Acceptance letter]

8 Apr 2021

PCOMPBIOL-D-20-01856R1 

WormPose: Image synthesis and convolutional networks for pose estimation in C. elegans

Dear Dr Stephens,

I am pleased to inform you that your manuscript has been formally accepted for publication in PLOS Computational Biology. Your manuscript is now with our production department and you will be notified of the publication date in due course.

With kind regards,

Andrea Szabo
